# A MicroRNA Next-Generation-Sequencing Discovery Assay (miND) for Genome-Scale Analysis and Absolute Quantitation of Circulating MicroRNA Biomarkers

**DOI:** 10.3390/ijms23031226

**Published:** 2022-01-22

**Authors:** Kseniya Khamina, Andreas B. Diendorfer, Susanna Skalicky, Moritz Weigl, Marianne Pultar, Teresa L. Krammer, Catharine Aquino Fournier, Amy L. Schofield, Carolin Otto, Aaron Thomas Smith, Nina Buchtele, Christian Schoergenhofer, Bernd Jilma, Bernhard J. H. Frank, Jochen G. Hofstaetter, Regina Grillari, Johannes Grillari, Klemens Ruprecht, Christopher E. Goldring, Hubert Rehrauer, Warren E. Glaab, Matthias Hackl

**Affiliations:** 1TAmiRNA GmbH, 1110 Vienna, Austria; kseniya.khamina@tamirna.com (K.K.); andreas.diendorfer@tamirna.com (A.B.D.); susanna.skalicky@tamirna.com (S.S.); moritz.weigl@tamirna.com (M.W.); marianne.pultar@tamirna.com (M.P.); teresa.krammer@tamirna.com (T.L.K.); 2Functional Genomics Center Zurich, ETH Zurich/University of Zurich, 8057 Zurich, Switzerland; catharine.aquino@fgcz.ethz.ch (C.A.F.); hubert.rehrauer@fgcz.ethz.ch (H.R.); 3MRC Centre for Drug Safety Science, Department of Molecular and Clinical Pharmacology, University of Liverpool, Liverpool L69 3GE, UK; A.L.Schofield@liverpool.ac.uk (A.L.S.); C.E.P.Goldring@liverpool.ac.uk (C.E.G.); 4Department of Neurology, Charité—Universitätsmedizin Berlin, 10117 Berlin, Germany; carolin.otto@charite.de (C.O.); klemens.ruprecht@charite.de (K.R.); 5Lilly Research Laboratories, Department of Investigative Toxicology, Non Clinical Safety Assessment and Pathology, Lilly Corporate Center, Indianapolis, IN 46285, USA; smith_aaron_thomas@lilly.com; 6Department of Clinical Pharmacology, Medical University of Vienna, 1090 Vienna, Austria; nina.buchtele@meduniwien.ac.at (N.B.); christian.schoergenhofer@meduniwien.ac.at (C.S.); bernd.jilma@meduniwien.ac.at (B.J.); 7Department of Medicine I, Medical University of Vienna, 1090 Vienna, Austria; 8Michael Ogon Laboratory for Orthopaedic Research, Orthopaedic Hospital Vienna-Speising, 1130 Vienna, Austria; Bernhard.Frank@khgh.at (B.J.H.F.); Jochen.Hofstaetter@oss.at (J.G.H.); 92nd Department, Orthopaedic Hospital Vienna-Speising, 1130 Vienna, Austria; 10Institute of Molecular Biotechnology, University of Natural Resources and Life Sciences, 1190 Vienna, Austria; Regina.Grillari@evercyte.com (R.G.); johannes.grillari@trauma.lbg.ac.at (J.G.); 11Evercyte GmbH, 1110 Vienna, Austria; 12Austrian Cluster for Tissue Regeneration, 1200 Vienna, Austria; 13Ludwig Boltzmann Institute for Traumatology, The Research Center in Cooperation with AUVA, 1200 Vienna, Austria; 14Department of Nonclinical Drug Safety, Merck Sharp and Dohme Corp., West Point, PA 19486, USA; warren_glaab@merck.com

**Keywords:** next-generation sequencing, small RNA-sequencing, microRNA, spike-in, biomarkers, toxicology, drug safety

## Abstract

The plasma levels of tissue-specific microRNAs can be used as diagnostic, disease severity and prognostic biomarkers for chronic and acute diseases and drug-induced injury. Thereby, the combination of diverse microRNAs into biomarker signatures using multivariate statistics seems especially powerful from the perspective of tissue and condition specific microRNA shedding into the plasma. Although next-generation sequencing (NGS) technology enables one to analyse circulating microRNAs on a genome-scale level, it suffers from potential biases (e.g., adapter ligation bias) and lacks absolute transcript quantitation as well as tailor-made quality controls. In order to develop a robust NGS discovery assay for genome-scale quantitation of circulating microRNAs, we first evaluated the sensitivity, repeatability and ligation bias of four commercially available small RNA library preparation protocols. The protocol from RealSeq Biosciences was selected based on its performance and usability and coupled with a novel panel of exogenous small RNA spike-in controls to enable quality control and absolute quantitation, thus ensuring comparability of data across independent NGS experiments. The established **mi**croRNA **N**ext-Generation-Sequencing **D**iscovery Assay (miND) was validated for its relative accuracy, precision, analytical measurement range and sequencing bias and was considered fit-for-purpose for microRNA biomarker discovery. Summarized, all these criteria were met, and thus, our analytical platform is considered fit-for-purpose for microRNA biomarker discovery from biofluids in the setting of any diagnostic, prognostic or patient stratification need. The established miND assay was tested on serum, cerebrospinal fluid (CSF), synovial fluid (SF) and extracellular vesicles (EV) extracted from cell culture medium of primary cells and proved its potential to be used across different sample types.

## 1. Introduction

MicroRNAs (miRNAs) are small endogenous non-coding RNAs of 17–25 nucleotides in length that regulate gene expression in mammalian cells. MicroRNAs are produced by virtually all cell types, and their expression can be changed in response to physiological stimuli and pathological processes. MicroRNAs can be measured in both tissues as well as liquid biopsies, including serum and plasma [1,2], and microRNA expression profiles in liquid biopsies differ between healthy and diseased individuals according to disease severity [3]. Furthermore, multiple studies indicated that levels of microRNAs can be used as diagnostic and prognostic biomarkers for various diseases including cardiovascular disease [4], infectious disease [5] and cancer [6,7]. Therefore, microRNAs have drawn much attention as promising diagnostic, disease severity and prognostic biomarkers [8,9].

The minimally invasive analysis of microRNAs in liquid biopsies allows for rapid, economical, and repeated sampling. This provides an opportunity for the development of screening programs and close monitoring of treatment response and disease progression [9]. The levels of tissue-specific microRNAs in liquid biopsies can be used for early detection of drug-induced organ injuries, such as miR-122 for liver [10], miR-133a for muscle, miR-124 for central nervous system [11], and miR-217 for pancreas [12] and offer a novel class of safety and toxicity biomarkers [13,14].

The Translational Safety Biomarker Pipeline (TransBioLine) Project of the IMI2 consortium aims to discover and qualify novel microRNA safety biomarkers for five organ systems (kidney, liver, pancreas, and vascular and central nervous systems) by 2024. Furthermore, TransBioLine aims to characterize inter- and intra-individual variability of circulating microRNAs through investigation of healthy volunteer populations. In order to achieve this goal, a robust assay for identification and quantitation of microRNAs shall be established.

Small RNA sequencing is a commonly used next-generation sequencing (NGS) technology for various types of short non-coding RNAs, including microRNAs in tissues and biofluids [15,16]. Several commercially available kits enable one to analyse circulating microRNAs in various tissues and body compartments. The NGS technology allowed the generation of a comprehensive map of the microRNA expression profiles across different sample types and conditions [17,18,19].

The detection of microRNA by NGS can be affected depending on the technical methods used for library preparation, which will bias the relative abundance of the selected microRNAs across different samples [20,21,22]. This bias results in over- or under-representation of certain microRNAs in particular samples. Furthermore, the standard approach to normalisation generates microRNA data in relative terms such as reads per million genome-matching reads (RPM) [23]. This method for normalisation assumes that the proportion of microRNAs remains constant across different tissues and datasets. Consequently, such a relative normalisation approach can generate misleading results when relative abundances of microRNAs vary between different sample types (e.g., tissues and biofluids), as well as experimental groups and populations. In order to resolve this issue, absolute normalisation for small RNA sequencing shall be established. Nonetheless, the absolute quantitation and comparison of microRNA levels remains a challenging task. Since only a few comparisons of small RNA NGS library preparations from plasma [19] have been published, therefore, we here aimed to (1) identify a small RNA NGS protocol that fits our criteria for specificity, sensitivity and usability, (2) couple it with an exogenous microRNA spike-in strategy for absolute quantitation and quality control, and (3) perform a fit-for-purpose validation.

In order to reach these aims, we comparatively evaluated four commercial small RNA library preparation protocols in combination with a novel exogenous spike-in control (**Mi**croRNA **N**GS **D**ata Analysis (miND) spike-in) for microRNA biomarker identification and absolute quantitation in plasma. We then implemented the conversion of read counts to the absolute amounts (amol or molecules/µL) in our analysis and selected the NGS protocol that fulfilled defined criteria including sensitivity, consistency, and ac-curacy of microRNA quantitation. The selected protocol was further optimized and subsequently validated for relative accuracy, analytical measurement range, precision and ligation bias. Finally, the established miND assay was tested in serum, cerebrospinal fluid (CSF), synovial fluid (SF) and extracellular vesicles (EV) extracted from cell culture media of primary cells.

## 2. Results

### 2.1. Study Design, Selection of Commercial Kits, and Reference Material

In order to evaluate the performance of small RNA sequencing library preparation protocols, we executed a systematic comparison of commercially available kits for small RNA sequencing and selected four kits for evaluation: QIAseq miRNA library kit (Qiagen, Hilden, Germany), RealSeq-Biofluids Plasma/Serum miRNA library kit (RealSeq Biosciences, Santa Cruz, USA), NEXTFLEX small RNA-seq kit v3 (Perkin Elmer, Waltham, MA, USA) and CleanTag small RNA library prep kit (TriLink Biotechnologies, San Diego, CA, USA) (Figure 1A, Appendix A). These kits were selected based on their complementary approaches to mitigate the risk of microRNA sequencing biases.

Three of the selected kits, CleanTag, NEXTFLEX and QIAseq, relied on sequential 5′ and 3′ adapter ligation to microRNAs followed by reverse transcription and PCR amplification. Two of these kits, NEXTFLEX and QIAseq, used random nucleotide sequences that serve as unique molecular indices (UMIs). UMIs enable the removal of duplicated reads introduced during amplification and increase the precision especially for low RNA input samples that require extensive library amplification [24]. The QIAseq kit used reverse transcription primer that contained an integrated UMI, while NEXTFLEX introduced random sequences within the adapters. The latter approach was suggested to decrease the sequence-specific ligation bias [20]. The RealSeq kit enabled ligation of microRNAs to a single adapter followed by the intramolecular circularization of the ligation product. This approach was shown to reduce the ligation bias by substituting the step of inter-molecular ligation between 5′ adaptor and microRNA by intramolecular circularization (Appendix A) [25].

Two types of reference samples were selected for the kit evaluation: RNA isolated from a pool of platelet-poor plasma (PPP) samples obtained from normal healthy volunteers (NHVs) and miRXplore Universal Reference (Miltenyi Biotec, Bergisch Gladbach, Germany), a synthetic miRNA reference pool comprising 1006 synthetic RNA oligonucleotides in equimolar concentrations matching mature microRNAs annotated in miRBase 9.2 database. PPP samples were used since they represent high quality biological samples [26] that are intended to be analysed within the IMI TransBioLine Project and enabled us to test the sensitivity of the selected assays. The miRXplore provided an opportunity to characterize the ligation bias for each small RNA sequencing protocol. Both sample types were used to evaluate the sensitivity and repeatability of the NGS protocol. miRXplore and plasma samples were analysed in 6 replicates starting from independent library preparation each (Figure 1A), meaning that a total of 48 samples were processed according to the selected protocols.

On the basis of these data, we aimed to select the commercial protocol with the best performance in order to establish absolute quantitation of microRNAs using a set of exogenous spike-in RNAs (Figure 1B).

### 2.2. Characterization of Four Small RNA Sequencing Protocols and Selection of the Protocol for the miND Assay

In order to evaluate the sequencing bias of the selected protocols, the synthetic equimolar pool of 1006 microRNAs, miRXplore Universal Reference, and the pool of plasma samples were used. First, we compared the sets of microRNAs that were identified by each protocol and assessed their relative values (RPMs). The results of unsupervised clustering analysis demonstrated that the samples were grouped by sample type as well as library preparation protocols (Figure 2A).

Although equimolar pooling was performed, the number of absolute reads varied between the NGS protocols. Absolute read counts for the miRXplore samples varied from 10 million to 60 million reads (avg = 22.2 million reads) between the samples and NGS protocols (Appendix A), but the relative proportion of the reads remained very consistent across the samples (Appendix A), with over 90% of reads mapped to miRXplore Universal Reference sequences. For plasma, the absolute number of microRNA-mapped reads varied between the protocols (Appendix A), while the relative number of miRNA reads was comparable (Appendix A).

None of the tested protocols was able to detect all microRNAs that were present in the miRXplore Universal Reference samples. The average number of the detected unique microRNAs varied between the kits, ranging from 501 for CleanTag to 616 for NEXTFLEX (RPM > 0) (Appendix A). The variations in the number of unique microRNAs detected by different NGS library preparation protocols were greater for plasma samples, ranging from 489 for the QIAseq kit to 818 for the RealSeq protocol (Appendix A).

Next, we characterized the sequencing bias of each protocol. We calculated the average RPM value (*n* = 6) detected for each microRNA in the miRXplore Universal Reference samples and expressed it as the relative fold change in comparison to the median RPM for each protocol (Figure 2B). In order to compare the protocols, we calculated the percentage of microRNAs within an 100× range (0.1- to 10-fold) of the median RPM. CleanTag showed the biggest variation in ligation efficiency (only 61.1% within the 100× range), while NEXTFLEX showed the lowest variation (80% within 100× range). To compare ligation-bias between protocols, we calculated Pearson correlation coefficients using the relative enrichment values for each microRNA and protocol and found that the two methods that use UMI for the mitigation of the sequencing bias, QIAseq and NEXTFLEX, demonstrated the highest correlation (PCC = 0.47) between the levels of detected microRNAs (Figure 2C and Appendix A).

To compare the sets of microRNAs that were identified by the selected protocols, a list of the 10 most and least abundant microRNAs detected with RPM > 10 was generated. The 10 most over-represented and under-represented sequences identified in the miRXplore Universal Reference samples showed no overlap between all four protocols (Appendix A). In plasma, only one microRNA, hsa-miR-486-5p, was consistently identified as one of the most abundant microRNAs by all four protocols (Appendix A); however, no overlap was found between the 10 least abundant microRNAs (Appendix A).

The overlap between the distinct microRNAs that were identified by each protocol was higher for miRXplore Universal Reference samples (Figure 2D) compared to the plasma samples (Figure 2E). In order to evaluate the consistency of each protocol in detecting endogenous microRNAs, we calculated the percentage of endogenous microRNAs that were only detected in one out of six technical replicates and used these values as a measure of inconsistency (Appendix A). CleanTag demonstrated the highest inconsistency (up to 40%) between the replicates for the miRXplore Universal Reference samples (Appendix A). For plasma, the percentage of inconsistently detected microRNAs also varied between the protocols and was lowest (<2%) for the RealSeq protocol (Appendix A).

In summary, the RealSeq protocol was selected since it returned the highest number of distinct microRNAs in plasma samples (Appendix A). Furthermore, the RealSeq protocol showed the lowest percentage of inconsistently detected microRNAs across the technical replicates for plasma samples (Appendix A), and acceptable ligation bias (Figure 2B). Finally, the RealSeq protocol was ranked high in terms of usability criteria such as cost, time of library preparation, convenience of use and consistency of performance. Based on the sensitivity, consistency of detection of microRNAs, and ligation bias as well as usability criteria, RealSeq was selected for further development of the small RNA sequencing pipeline for absolute quantitation.

### 2.3. Design and Testing of the miND Spike-In for Quality Control and Absolute Normalisation of Small RNA Sequencing Data

Quality control and absolute normalisation of small RNA sequencing data require defined standards that remain constant across different conditions and experiments. Therefore, we conceived the following workflow: we designed 227 exogenous oligonucleotides (“spike-ins”), each being a 21 nucleotide (nt) long RNA, based on the methodology that was published by Lutzmayer et al., 2017 [27]. Each spike-in consisted of a 13-nucleotide core sequence that was flanked by four randomized nucleotides on the 5′ and 3′ ends. Next, we mapped our list of 227 candidates against 5 mammalian genomes (*Homo sapiens*, *Mus musculus*, *Bos taurus*, *Rattus norvegicus*, *Sus scrofa*) allowing for one mismatch. We selected seven core sequences that demonstrated the lowest overlap with the selected genomes. Each core sequence was flanked by four randomized nucleotides on the 5′ and 3′ ends (Table 1), resulting in 65,536 different oligonucleotides per spike-in. We hypothesised that the ligation bias of each spike-in set with the same core sequence would be minimized due to the presence of random nucleotides on the 5′ and 3′ ends of the miND spike-in [21].

Spike-ins were then mixed in specific molar ratios to cover the dynamic range of endogenous microRNA concentrations in human plasma samples and diluted into working stocks, which were added to the RNA samples before library preparation and processed with the RealSeq protocol (see workflow in Figure 1B). Since it was shown before that the choice of RNA extraction protocol can impact the detection of small RNA in plasma [28], we set out to evaluate the performance characteristics of two RNA extraction methods in combination with the miND spike-ins and the RealSeq protocol: miRNeasy Mini kit (Qiagen, column-based) and Maxwell RSC miRNA Tissue kit (Promega, bead-based). This experiment allowed us to determine (1) the performance of the miND spike-in in combination with the RealSeq NGS protocol, and (2) the compatibility of column- and bead-based RNA extraction protocols with the designed workflow.

To evaluate the performance of miND spike-ins, we assessed the relation between attomolar concentrations of the miND spike-ins and the observed read counts using a linear model without intercept. This model was further used to predict the absolute concentrations of endogenous microRNA molecules per μL of the input RNA.

The miND spike-in sequences were identified in all samples extracted both with Qiagen and Promega protocols (*n* = 3 per RNA extraction protocol), and miND spike-in counts (RC) and absolute concentrations (molecules/μL) demonstrated high correlation (Pearson’s r value > 0.99 for all samples) (Figure 3A and Appendix A).

The miND spike-ins showed compatibility with both RNA extraction protocols, as the measurement range covered a dynamic range of 5 log10 s for observed RPM values and showed good overlap with read counts from endogenous microRNAs (Figure 3B). The highly linear relationship between the observed read counts and absolute concentrations of the miND spike-ins in both RNA extraction protocols (Figure 3A) as well as the overlap between endogenous microRNA counts and spike-in counts (Figure 3B) confirmed that the selected spike-in range was suitable for analysis of the majority of endogenous microRNAs in plasma samples independent of the RNA extraction protocol.

Next, we used a linear regression model to calculate absolute concentrations of endogenous microRNAs (Figure 3C) for each RNA extraction protocol and observed a statistically significantly lower number of microRNA molecules per microliter for the Promega RNA isolation protocol compared to the Qiagen protocol (unpaired *t*-test, *p*-value < 0.0001). This difference in microRNA abundance was not detectable using relative normalisation (RPM data), emphasizing the advantages of absolute normalisation. The lower RNA extraction efficiency was confirmed by RT-qPCR analyses of 13 low to high abundant microRNAs, which showed that Cq-values from Promega-extracted samples were on average 1.61 Cq-units higher compared to the values from Qiagen-extracted samples (Appendix A).

Another important factor is the depth of sequencing that can introduce variability in the identification of microRNAs [29]. In order to reveal how the depth of sequencing affects the number of microRNAs detected with our assay, we pooled the obtained reads for each RNA extraction protocol and sub-sampled them five times to achieve 30, 15, 7.5, 3.75 and 1.875 million reads per sample (five technical replicates per condition). These sub-sampled files were subjected to bioinformatic analysis, and the number of distinct microRNAs identified for each condition was obtained (Appendix A). In line with the lower absolute concentrations of microRNAs obtained with the Promega kit, we observed that the Promega kit consistently delivered a lower number of microRNAs compared to the Qiagen kit. The generated dataset demonstrated how an increase in the sequencing depth affects the number of detected microRNAs (Appendix A), can compensate for lower RNA extraction efficiency, and can serve to develop a guideline for selection of sequencing platforms and multiplexing strategies for future projects.

We concluded that the miND spike-in was successfully used for absolute normalisation in the samples extracted with both the Qiagen and Promega protocols. Despite the fact that the number of detected endogenous microRNAs at the same depth of sequencing was higher in Qiagen-extracted samples compared to the Promega-extracted samples (Appendix A), the miND spike-in generated a nearly-perfect linear model in all samples (Appendix A). Considering the fact that the RNA extraction protocol from Qiagen allows the processing of 12 samples in parallel, while the Promega protocol allows 48, we decided to use the latter approach for our assay. Furthermore, the Promega protocol still enabled us to consistently detect the most abundant microRNAs.

### 2.4. Fit-for-Purpose Validation of the Established NGS Protocol

In order to validate the established miND assay, we characterized relative accuracy, precision, analytical measurement range and sequencing bias.

Relative accuracy of the miND assay was determined by analysing two plasma pools obtained from normal healthy volunteers (NHVs) and patients with acetaminophen-induced liver injury (APAP). The APAP samples served as an example of patient samples with perturbed microRNA expression profiles [30].

The plasma pools of NHV and APAP samples were either directly analysed, or APAP plasma was spiked into the NHV samples in the following proportions: 1:1, 1:5 and 1:10 in triplicates. This approach allowed us to simultaneously determine the titration response of multiple microRNAs, thus providing an opportunity to draw a reliable conclusion about the relative accuracy of the assay for different microRNAs.

The differential gene expression analyses identified 266 differentially regulated microRNAs (FDR < 0.05) between NHV and APAP samples (Appendix A). Thirty-seven microRNAs that were up-regulated in APAP samples by more than 10-fold compared to NHV samples were selected for analysis of relative accuracy by comparing the observed to the expected values obtained for 1:1, 1:5, and 1:10 dilutions from relative (RPM) and absolute (aMol) normalised datasets (Figure 4A,B). The linear correlation between observed and expected values for absolute concentrations as well as RPMs of 37 microRNAs with varying baseline levels was investigated. Calculated Pearson correlation coefficients were transformed to z-score by Fisher transformation and resulted in z = 0.649 for the data generated based on the absolute concentrations and z = 0.556 for the RPM-based data. On average, 30 out of 37 investigated microRNAs demonstrated a percentage of recovery between 50 and 200%. The obtained results demonstrate the potential of absolute normalisation based on the miND spike-in to reveal precise and accurate microRNA concentrations across different samples.

Next, reproducibility and repeatability of the established NGS protocol were evaluated. We used a pool of RNA isolated from NHV plasma samples that was processed by three operators on three independent days. Thus, a total of nine libraries were generated to assess intra-operator repeatability and inter-operator reproducibility. We found that the coefficient of variation (CV%) across all nine replicates was associated with the mean concentration (Figure 4C). Repeatability was estimated based on the portion of microRNAs with CV < 50% between the technical replicates for each operator, and reproducibility was calculated for all nine technical replicates processed by three operators. For three operators, over 93% of microRNAs demonstrated CV < 50% (Appendix A) and for all nine technical replicates, this value was 59% (Figure 4C).

The analytical measurement range of the NGS protocol was defined by concentrations of the highest (I, 50 amol) and the lowest (E, 0.005 amol) miND spike-ins. The microRNA profile data, which were generated to evaluate reproducibility and repeatability of the NGS protocol, were used to investigate the average portion of endogenous microRNAs within the analytical measurement range. The miND spike-in E with the lowest concentration was detected in eight out of nine technical replicates. On average, 69.56% of the detectable microRNAs (RPM > 0) were within the analytical measurement range as defined by the miND spike-in (Figure 4D).

Next, the sequencing bias of the established NGS protocol was investigated, since the analysis of miRXplore Universal Reference indicated that the RealSeq as well as other tested protocols for small RNA sequencing exhibited sequencing bias (Figure 2B). This bias can result in the over- or under-representation of microRNAs in small RNA sequencing datasets. To evaluate the effect of bias on the accuracy of our NGS protocol, we selected three microRNAs that were previously detected in the miRXplore samples to be either under-represented (hsa-miR-137-3p), over-represented (hsa-miR-630) or normally represented (hsa-miR-520e-3p) using the miRXplore samples (Appendix A) and that were detected using the optimized NGS protocol in plasma samples as well. These microRNAs were synthesized as RNA oligonucleotides and spiked into the pool of RNA isolated from NHV at defined molar amounts (5 amol, 20 amol, 80 amol).

The calculated increase in concentration that was detected in the spiked-in samples demonstrated variations between the selected microRNAs (Figure 4E). The microRNA that was under-represented in the miRXplore Universal Reference dataset, hsa-miR-137-3p, showed a significantly lower increase in the spiked concentration compared to the expected one. Even in the sample, which contained 80 amol of the synthetic spike-in, the detected concentration for hsa-miR-137-3p was only 21 amol. In contrast, two microRNAs that were present at the expected level or over-represented in the miRXplore Universal Reference dataset, hsa-miR-520e-3p and hsa-miR-630, showed higher increases in the absolute concentrations compared to the expected ones. These results underline the importance of understanding sequencing biases of the applied small RNA NGS workflow for the profiling of microRNAs and point out that different microRNAs might be affected in different ways.

The obtained results allowed us to validate the relative accuracy, precision, and analytical measurement range. Furthermore, it enabled us to characterize the sequencing bias of the established NGS protocol. A better understanding of these parameters of the miND assay ensures the correct interpretation of the obtained results.

### 2.5. NGS Analyses of Diverse Biological Samples with the miND Spike-In Assay

In order to evaluate the performance of the miND spike-in in different types of biological samples, we analysed plasma, serum, cerebrospinal fluid (CSF), synovial fluid (SF) and extracellular vesicles (EV), which were isolated from cell culture medium of primary human cells in triplicates.

All miND spike-in sequences were detected in each of the analysed samples. Furthermore, the selected range of the miND spike-ins covered the range of endogenous microRNAs (Figure 5A). Next, we calculated RPM (Figure 5B) as well as the number of microRNA molecules/µL using miND spike-ins (Figure 5C, Appendix A). It was observed that the transformation to copy numbers allowed a better interpretation of the differences in miRNA concentrations between sample types, i.e., highest concentrations for serum and lowest for CSF. This analysis also demonstrated the high range of microRNA concentrations observed in biofluids, with few high abundant miRNAs and a large number of low abundant miRNAs between 1 and 10 copies/µL biofluid.

Finally, the established miND spike-in assay also allowed us to estimate the mean number of microRNA molecules detected in 1 µL of biofluid by adjusting to the biofluid input volumes (Table 2).

This experiment demonstrated the broad application spectrum of miND spike-ins for absolute quantitation of small RNAs in various sample types, specifically samples with low RNA input, such as biofluids and EVs. Importantly, the miND spike-ins enabled us to directly compare the absolute abundances of microRNAs extracted from different sample types and improve our understanding of microRNA compositions in different sample types.

## 3. Discussion

Here, we reported on the development and characterization of a microRNA Next-Generation-Sequencing Discovery Assay (miND) for the identification of circulating microRNA biomarkers in liquid biopsies using absolute quantitation. Based on the methodology that was developed by Lutzmayer et al. [27], we generated a set of exogenous small RNA spike-in controls containing 13 nt core sequences that were flanked by a set of four randomized nucleotides on the 5′ and 3′ ends. This design strategy provides up to 65,536 possible sequences for each of the core sequences. The randomized nucleotides are expected to minimize potential sequencing biases that might be caused by ligation and amplification biases. While Lutzmayer et al. developed and characterized the spike-ins for small RNA sequencing in plant tissues, we designed the miND spike-ins for human, rodent, as well as other mammalian species, and validated the workflow for human plasma samples.

Our results suggest that the miND assay allows users to determine absolute concentrations of microRNAs in total RNA samples obtained from plasma as well as other liquid biopsies (serum, CSF, SF) and, therefore, enables the comparison of data within or across sample types irrespective of differences in RNA composition that would affect relative normalisation.

To achieve this, we first performed a systematic comparison of four commercially available small RNA library preparation protocols in terms of sensitivity, consistency, and ligation bias (Figure 2 and Appendix A). The kits were selected based on their complementary strategies to reduce sequencing bias. Based on its sensitivity and consistency, we decided to work with the RealSeq protocol as the basis for further assay characterization and implementation of absolute quantitation. For this, in-silico analyses were performed to define seven exogenous 13-mer core-nucleotide sequences as the basis for a set of spike-in calibrators. We tested the performance of the spike-in calibrator in combination with two different RNA extraction protocols (precipitation/column-based vs. bead-based) and observed a nearly perfect correlation between the detected RC and amount of added miND spike-ins (molecules/µL) (Figure 3A and Appendix A). Finally, we validated relative accuracy, precision, analytical measurement range, and sequencing bias for the final workflow and demonstrated compatibility of miND spike-ins with serum, CSF, synovial fluid and EVs, in addition to plasma.

The results obtained during the development of this workflow clearly show the impact that individual sample preparation steps can have on the final NGS data. For instance, different library preparation protocols resulted in different sequencing biases (Figure 2B,C) and detected different sets of microRNAs both in miRXplore (Figure 2D) and in plasma (Figure 2E) samples. In addition, sequencing bias can result in the over- or under-estimation of microRNA effects (Figure 4E), which should be considered during the selection of biomarker candidates. Finally, the choice of the RNA extraction protocol (Appendix A) as well as sequencing depth (Appendix A) impacts the assay sensitivity and should, therefore, be consciously selected for each individual project.

### The Value of Spike-Ins for Small RNA Sequencing

The use of spike-in controls to monitor data quality is a common strategy for messenger RNA (mRNA) sequencing [31,32] but has not yet been adopted for small RNA [33] sequencing workflows. Therefore, there is no accepted standard to compare and perform quality control of small RNA sequencing data other than assessing sequence quality, read composition, or sequencing depth. Given the constant improvement or development of new small RNA sequencing protocols, such a standard would be highly desirable. Besides quality control, spike-ins could also satisfy the unmet need for alternative NGS normalisation strategies. The commonly used relative normalisation as reads per million total (or microRNA) reads (RPMs) assumes that the overall amount of microRNA or total RNA composition is constant across all samples in an experiment. However, it must be assumed that this is not the case when comparing between liquid biopsy sample types with very specific RNA compositions [19] or between different individuals [34]. Lutzmayer et al. recognized this need in the context of plant microRNA research [33,35], but the adaptation of the technology to mammalian species as well as liquid biopsies was still missing.

In order to meet this need, we designed and developed the miND spike-ins and demonstrated their compatibility with a commercial small RNA sequencing protocol. The broad concentration range that covers a significant range of concentrations of the endogenous microRNAs enables the application of the miND spike-in with various sample types that exhibit different microRNA concentration ranges (Table 2). By performing absolute quantitation, we were able to improve significantly the relative accuracy compared to RPM-based normalisation (Figure 4A,B). However, further studies will be required using clinical samples from control and disease cohorts to determine the value of miND spike-ins for relative comparison of microRNA levels.

Most importantly, however, absolute quantitation is required to obtain a realistic view of the true amounts of microRNA present in a given biological sample. For example, it allowed us to confirm RT-qPCR results that suggested lower microRNA extraction efficiency for a bead-based protocol compared to precipitation- and column-based RNA purification also by NGS (Figure 3C and Appendix A), which was not evident from relative normalisation using RPM values. Moreover, from a clinical point of view, there are clear advantages of this approach: by analysing four different biofluids (serum, plasma, CSF, synovial fluid) we were able to provide an accurate estimation of microRNA abundance. Especially in the context of CNS injury, it was interesting to observe that CNS enriched miRNAs such as miR-124-3p show 50- to 100-fold higher concentrations in CSF than in plasma. This result confirms their potential as biomarkers of chronic or acute brain injury, since increased blood–brain barrier permeability could result in a measurable increase in peripheral blood levels of these miRNAs.

To verify the utility of this NGS workflow and miND spike-ins, their incorporation into the analysis of larger sample cohorts will be required, and benchmarking against relative normalisation strategies needs to be performed. Nevertheless, our results demonstrate the high potential of spike-ins for quality control of small RNA sequencing experiments and absolute quantitation of microRNAs across different sample types, biological conditions and datasets. Furthermore, the absolute quantitation of circulating microRNAs detected by the miND assay will be leveraged in the TransBioLine project in order to identify sensitive and specific microRNA biomarkers for individual disease states or drug-induced injuries.

## 4. Materials and Methods

### 4.1. Samples

Platelet-poor plasma (PPP) samples from 16 healthy individuals were collected and pooled. The samples were obtained from the placebo period of a clinical study investigating the effects of desmopressin on endothelial activation [36]. The independent Ethics Committee of the Medical University of Vienna (1178/2018) and the competent authorities (Austrian Agency for Health and Food Safety) approved the study, which was conducted in accordance with the Declaration of Helsinki and the Good Clinical Practice guideline. All subjects provided informed consent prior to any trial-related activities. Citrate-anticoagulated blood samples were obtained by fresh venepuncture and immediately put on ice, and platelet poor plasma was generated by two centrifugation steps (2000× *g* for 10 min and 10,000× *g* for 10 min at 4 °C). The samples were stored at −80 °C.

The miRXplore Universal Reference 25 (Cat. 130-093-521, Miltenyi Biotec, Bergisch Gladbach, Germany), the reference pool comprising more than 950 single-stranded synthetic RNA oligonucleotides in equimolar concentrations matching mature microRNAs annotated in the miRBase 9.2 sequence database, was used. The lyophilized oligonucleotides were dissolved in 28 μL sterile RNase-free water and stored at −80 °C.

Usage of −80 °C stored serum and CSF samples for the purposes of this study was approved by the ethics committee of Charité—Universitätsmedizin Berlin (EA1/258/19). Written informed consent was obtained from all subjects.

Synovial fluid samples were obtained in accordance with the ethical guidelines of the Declaration of Helsinki. This study was approved by the institutional ethics committee of the Vinzenz Group (registration number: EK10/2020). Informed consent was obtained from all subjects. The samples were stored at −80 °C.

Primary human cells were cultivated in DMEM (Gibco, Franklin, MA, USA) supplemented with 10% foetal bovine serum (FBS) and 4 mM L-Glutamine (Gibco, MA, USA). For generating cell conditioned medium, cells were incubated serum-free for 24 h using DMEM supplemented with 4 mM L-Glutamine. Sixty-five millilitres of conditioned medium was harvested and sequentially centrifuged for 10 min each at 200× *g* and 2000× *g*, respectively. After the centrifugation, the medium was filtered through a 0.45 µM filter and concentrated to a volume of around 2.5 mL using Amicon Ultra-15 30 kDa filters (Millipore Sigma, Darmstadt, Germany). Two millilitres of the concentrated conditioned medium was applied to a qEV2 35 nm size-exclusion chromatography column (IZON, Medford, MA, USA). Fractions containing extracellular vesicles were collected according to the manufacturer’s recommendation, concentrated using Amicon Ultra-15 10 kDa filters and used for isolation of RNA. The samples were stored at −80 °C.

### 4.2. Design of the miND Spike-Ins

The miND spike-ins were designed following the protocols described by Lutzmayer et al. [28].

The spike-ins were synthesized in collaboration with Integrated DNA Technologies (IDT). The lyophilized oligonucleotides were dissolved in sterile RNase-free water to 100 μM. Next, the oligonucleotides were diluted and mixed to obtain the miND spike-in solution containing corresponding attomolar concentrations for each spike-in (Table 1)

### 4.3. RNA Extraction

Total RNA was extracted from plasma, serum, SF, and CSF samples using the miRNeasy Mini kit following the manufacturer’s protocol (Cat. 217004, Qiagen). Briefly, aliquots of 200 μL were lysed in QIAzol Lysis reagent, followed by incubation with chloroform. After the phase separation, the upper aqueous phase was mixed with glycogen (final concentration 50 μg/mL) and subjected to automated RNA purification in a QIAcube. Total RNA was eluted in 30 μL of nuclease-free water and stored at −80 °C.

Total RNA was extracted from 200 μL of plasma samples with Maxwell RSC miRNA Tissue kit (Promega, Madison, WI, USA, AS1460) according to the manufacturer’s protocol. Briefly, samples were thawed on ice and centrifuged at 12,000× *g* for 5 min to remove cellular debris. For each sample, 200 μL of plasma were consequently mixed with the following reagents: 200 μL of homogenization solution, 200 μL of lysis buffer and 15 μL of Proteinase K. Next, samples were incubated for either 15 min or 2 h on a heat block at 37 °C, 300 rpm. After the incubation, samples were transferred to RSC cartridges followed by automated RNA extraction with the Maxwell instrument. Finally, total RNA was eluted in 50 μL nuclease-free water and stored at −80 °C prior to further analyses.

### 4.4. NGS Library Preparation

The RNA samples isolated from the plasma pool and miRXplore Universal reference were used for small RNA library preparation with four kits: NEXTFLEX Small RNA-Seq kit v3 for Illumina platforms (Cat. NOVA-5132-06, PerkinElmer, Waltham, MA, USA), QIAseq miRNA Library kit (Cat. 331502, Qiagen), CleanTag small RNA library prep kit (Cat. 040L-3206-24, TriLink Biotechnologies) and RealSeq-Biofluids NGS library preparation kit for miRNAs and small RNAs for total RNA samples from biofluids (Cat. 600-00012-SOM, RealSeq Biosciences, protocol 20181220_RealSeq-BF_CL). The maximal possible sample volume of plasma RNA was used as an input for each protocol. One microlitre of either nuclease-free water or miND spike-in were added to each reference sample before the library prep. One picomole of miRXplore Universal Reference was used for each library prep. The input of plasma RNA for library preparation was 9.5 μL for NEXTFLEX, 4 μL for QIAseq, 1 μL for CleanTag and 8.5 μL for RealSeq kits. The adapters were pre-diluted to account for low miRNA abundance in plasma samples.

The library preparation workflows were performed for NEXTFLEX, QIAseq, CleanTag and RealSeq kits according to the manufacturer’s protocols. For NEXTFLEX and RealSeq protocols, adapters were pre-diluted 1:4 and PCR amplification was performed for 23 cycles. For QIAseq, PCR adapters were pre-diluted 1:5 and amplification was performed for 23 cycles. For CleanTag, PCR adapters were pre-diluted 1:12 and amplification was performed for 22 cycles.

In total, 54 microRNA libraries were prepared and analysed for library fragments distribution on an Agilent DNA 1000 kit (Agilent Technologies, Santa Clara, CA, USA, 5067-1504) with Agilent DNA1000 reagents (Agilent Technologies, 5067-1505).

The generated libraries were pooled in equimolar proportion, and the obtained pool was size-selected with the BluePippin system using 3% agarose cassette, 100–250 kb (Sage Science, Beverly, MA, USA, BDQ3010) to remove DNA fragments outside of the target range. The pooled and purified libraries were analysed for fragment distribution with the Agilent High Sensitivity DNA kit (Agilent Technologies, 5067-4626) with Agilent High Sensitivity DNA reagents (Agilent Technologies, 5067-4627).

The library pool was sequenced on an Illumina NextSeq550 (single-read, 75 bp) according to the manufacturer’s protocol.

### 4.5. RT-qPCR

cDNA was synthesized from total RNA using the miRCURY LNA RT kit (Qiagen, 339340). The samples were processed according to the manufacturer’s specifications. In total, 2 μL of total RNA were used per 10 μL reverse transcription (RT) reaction. PCR amplification was performed using a 96-well plate format on a Roche LC96 instrument (Roche Diagnostic) using miRCURY LNA SYBR Green PCR kit (Qiagen, 339347) with the following settings: 95 °C for 2 min, 45 cycles of 95 °C for 10 s, and 56 °C for 60 s, followed by melting curve analysis. To calculate the cycle of quantitation values (Cq-values), a combination of the 2nd derivative maximum and the fits point method (LC96, Roche software) was used. Cel-miR-39 was added to the total RNA sample prior to reverse transcription and qPCR to measure the efficiency of cDNA synthesis.

### 4.6. Bioinformatic Analyses

The bioinformatic analyses were performed with the software package miND, a data analysis pipeline that generates overall QC data, unsupervised clustering analysis, normalised microRNA count matrices, and differential expression analysis based on raw NGS data. Overall quality of the next-generation sequencing data was evaluated automatically and manually with fastQC v0.11.8 [37] and multiQC v1.7 [38]. Reads from all passing samples were adapter trimmed and quality filtered using cutadapt v2.3 [39] and filtered for a minimum length of 17 nt. Mapping steps were performed with bowtie v1.2.2 [40] and miRDeep2 v2.0.1.2 [41], whereas reads were mapped first against the genomic reference GRCh38.p12 provided by Ensembl [42], allowing for two mismatches, and subsequently miRBase v22.1. [43], filtered for microRNAs of hsa only, allowing for one mismatch. For a general RNA composition overview, non-microRNA mapped reads were mapped against RNAcentral [44] and then assigned to various RNA species of interest. Statistical analysis of pre-processed NGS data was performed with R v3.6 and the packages pheatmap v1.0.12, pcaMethods v1.78 and genefilter v1.68. Differential expression analysis with edgeR v3.28 [45] used the quasi-likelihood negative binomial generalized log-linear model functions provided by the package. The independent filtering method of DESeq2 [46] was adapted for use with edgeR to remove low abundant microRNAs and thus optimize the false discovery rate (FDR) correction. The absolute normalisation was performed based on the predefined spike-in concentrations [28].

The subsampling was performed using Seqtk v1.3 tool [47].

### 4.7. Data Analyses and Statistical Methods

Data processing and analyses were conducted using Microsoft Excel version 16.57 [48] and GraphPad Prism version 9.1 (GraphPad Software, San Diego, CA, USA, www.graphpad.com (accessed on 13 December 2021).) [49]. Data were visualised using R v4.3 and the packages ggplot2, ggpubr and enhancedVolcano by generating scatter plots and Volcano plots. Graphics were created using BioRender.com [50].

## 5. Conclusions

We established and validated the miND assay that enables one to perform absolute normalisation of small RNA sequencing data based on the concentrations of exogenous spike-ins.

## 6. Patents

A patent was filed for the invention related to novel spike-in oligonucleotides for use in quantitative normalisation of nucleotide sequence data and was granted in the EU under EP3354746B1.

## Figures and Tables

**Figure 1 ijms-23-01226-f001:**
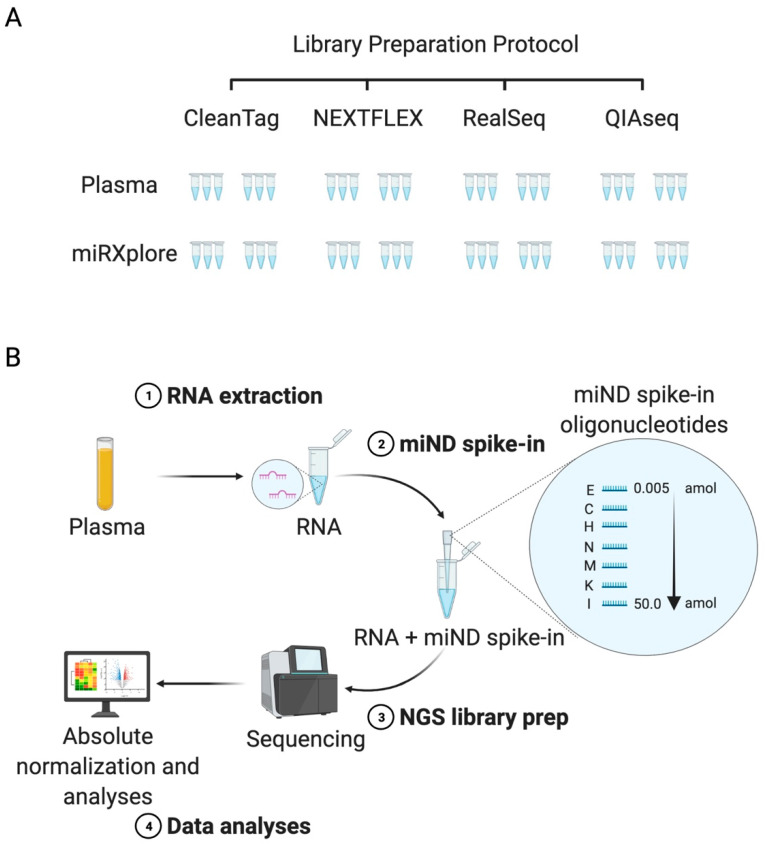
Study design. (**A**) Four library preparation protocols were selected in order to evaluate their performance and usability. Six technical replicates from a pool of plasma samples and aliquots of miRXplore Universal Reference were analysed. (**B**) The NGS assay utilizing mind spike-ins was designed using the following steps: (1) RNA extraction; (2) adding of the miND spike-in—a pool of 7 oligonucleotides, each containing a 13 nt core sequence flanked by a set of 4 randomized nucleotides on the 5′ and 3′ ends, that were mixed in the defined ratio; (3) preparation of the NGS libraries according to the selected protocols; (4) data analyses and data normalisation based on the miND spike-in concentration range.

**Figure 2 ijms-23-01226-f002:**
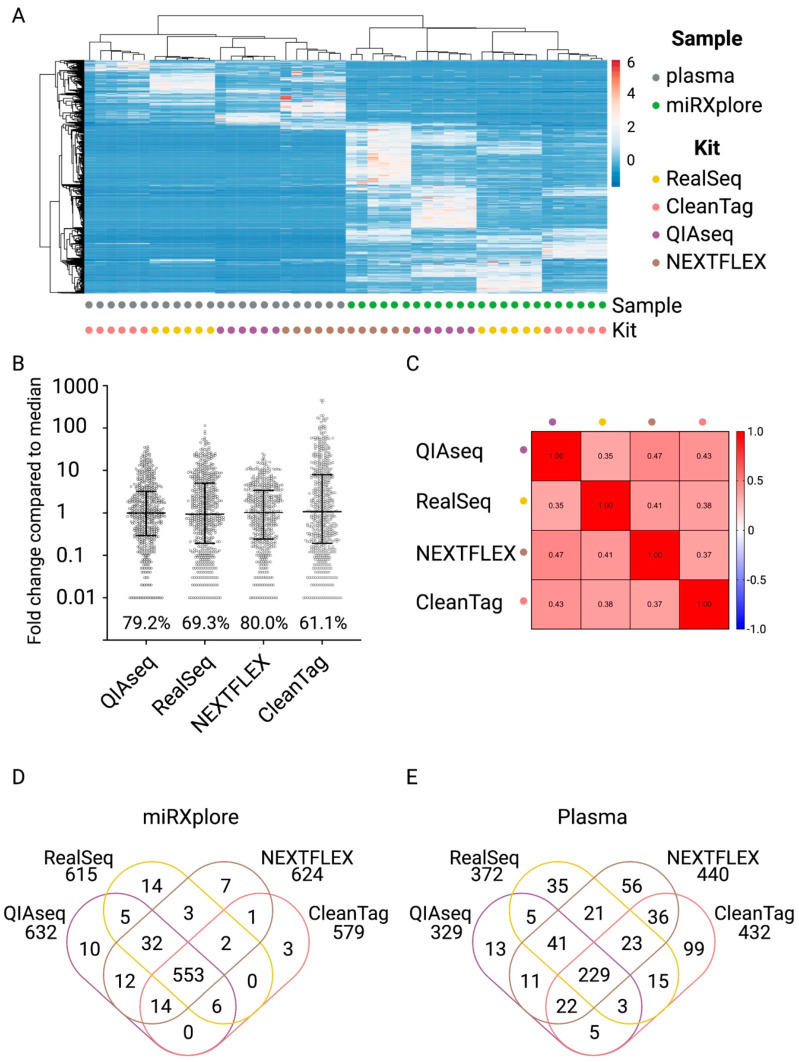
A systematic comparison of the selected small RNA library preparation protocols. (**A**) Unsupervised clustering analyses of plasma and miRXplore samples based on RPM data from 739 microRNAs. (**B**) Sequencing bias. RPM values were averaged across six technical replicates per protocol. The fold change (FC) compared to the median RPM was calculated for each microRNA and sample, and is presented on a log10 scale. Medians with interquartile range are indicated. The percentage of microRNAs within 0.1- to 10-fold from the median (=1) was calculated and depicted above the x-axis. (**C**) Pearson correlation coefficient matrix was generated from the data shown in (**B**). (**D**) Venn diagram representing the overlap between distinct microRNAs detected with four selected protocols in the miRXplore samples. (**E**) Venn diagram representing the overlap between distinct microRNAs detected with four selected protocols in the plasma samples.

**Figure 3 ijms-23-01226-f003:**
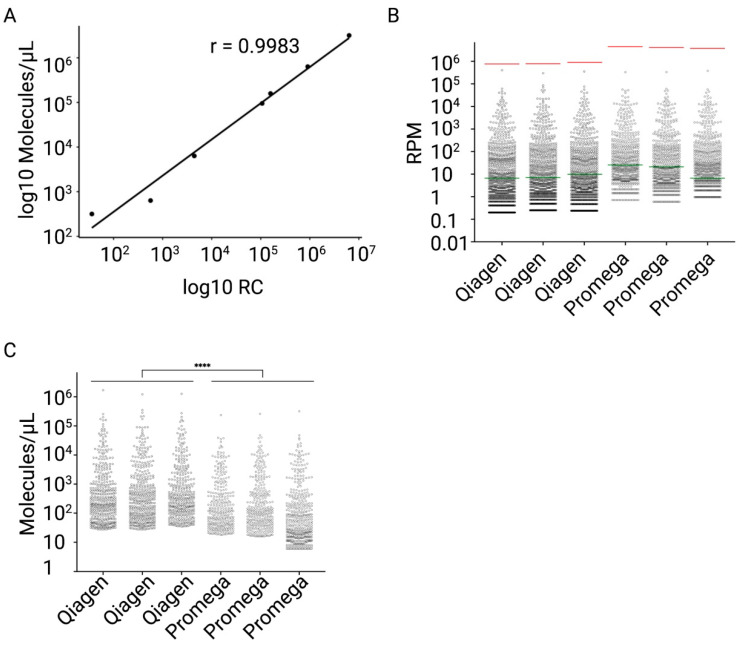
Testing of the miND spike-in. Aliquots of a plasma pool were processed either with Qiagen or Promega RNA extraction protocols in triplicates. The generated RNA samples were analysed with the miND assay. (**A**) Scatter plot of relative miND spike-in (RC) compared to absolute miND spike-in levels (molecules detected per microliter of RNA) in Promega-extracted sample (Replicate 01). Pearson correlation coefficient r as well as the line that represents a linear model derived from the plotted values were calculated. (**B**) Distribution of RPM values of the endogenous microRNAs in the Qiagen- and Promega-extracted samples (3 replicates per protocol). The miND spike-ins with the highest (I, 50 amol) and the lowest (E, 0.005 amol) concentrations were indicated with red and green lines on the plot. (**C**) Distribution of absolute concentrations (molecules per microliter of RNA) of the endogenous microRNAs in the Qiagen- and Promega-extracted samples (3 replicates per protocol). The average values of molecules/µL for each sample were calculated, and an unpaired t-test comparison for Qiagen- and Promega-extracted samples was performed (*p*-value < 0.0001, *p*-value summary ****).

**Figure 4 ijms-23-01226-f004:**
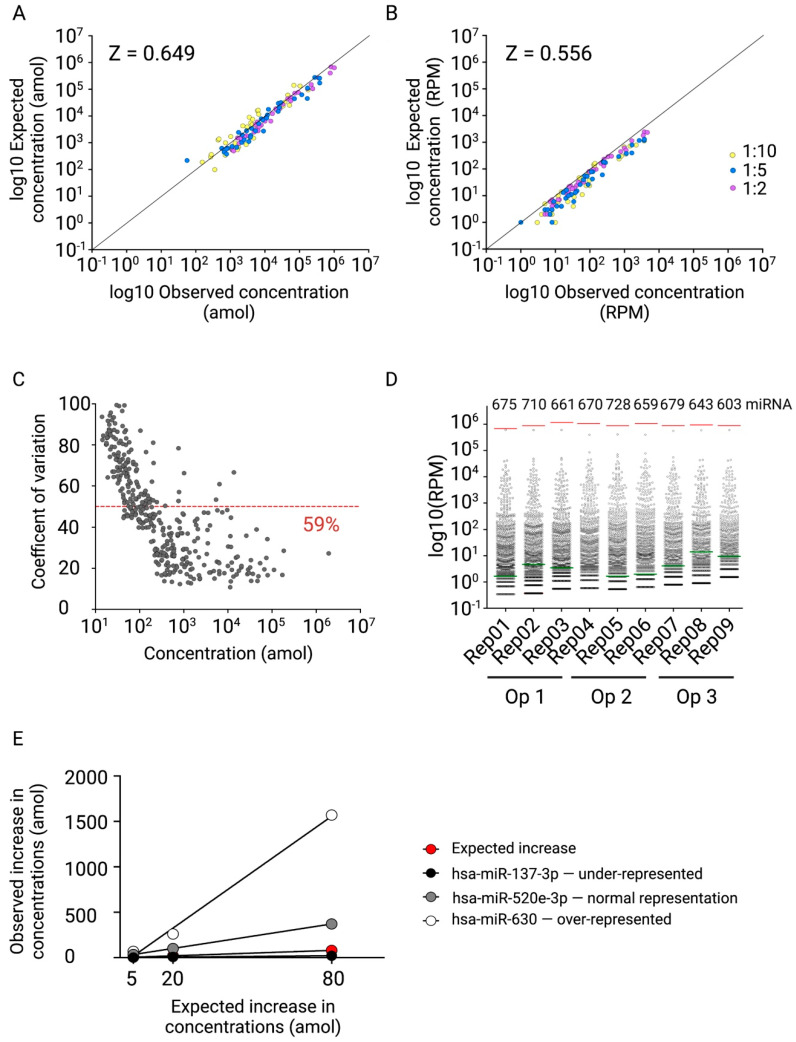
Observed vs. expected absolute concentrations in (**A**) amol and (**B**) RPM from 37 microRNAs that were significantly up-regulated in APAP compared to NHV samples, presented on log10 axes. The Pearson correlation coefficient of dilution series was calculated for 37 microRNAs and transformed to z-score using Fisher transformation. (**C**) The coefficient of variation for each microRNA from 9 technical replicates was calculated and plotted against the absolute concentrations (amol). The red line indicates the fraction of microRNAs with less than 50% CV. (**D**) Distribution of RPM values of the endogenous microRNAs for 9 technical replicates (RPM > 0). The miND spike-in with the highest (I, 50 amol) and the lowest (E, 0.005 amol) concentrations indicated with red and green lines. The number of detected microRNAs is indicated above each sample. (**E**) Observed vs. expected increase in absolute concentrations (amol) for hsa-miR-137-3p (under-represented), hsa-miR-520e-3p (normal representation) and hsa-miR-630 (over-represented). The synthetic microRNAs were spiked in the RNA isolated from a pool of NHV samples at 5, 20 and 80 amol concentrations.

**Figure 5 ijms-23-01226-f005:**
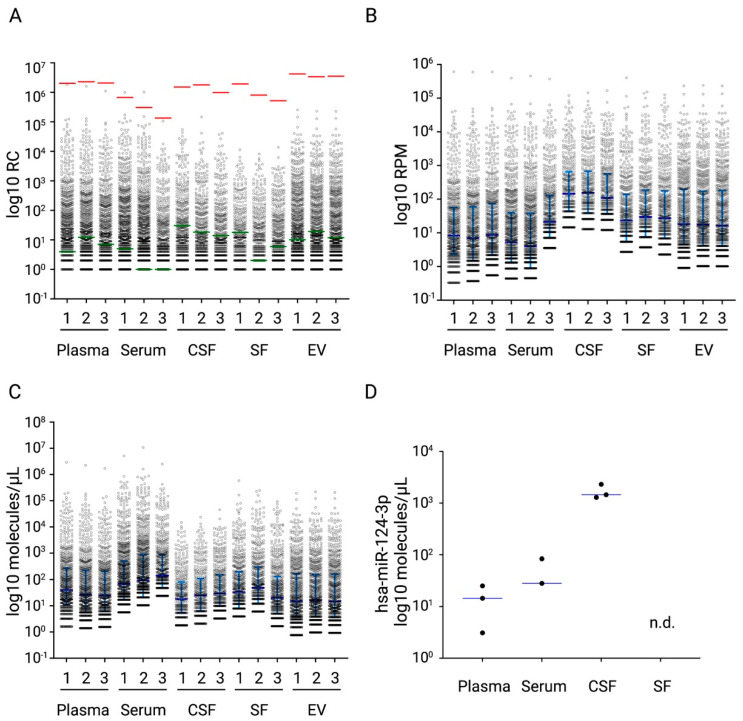
Distribution or read count, RPM values and concentrations for endogenous microRNA detected in 3 samples for plasma, serum, cerebrospinal fluid (CSF), synovial fluid (SF) and extracellular vesicles (EV) extracted from cell culture medium of primary human cells are presented on log10 scale. (**A**) microRNA RC values are demonstrated. The miND spike-ins with the highest (I, 50 amol) and the lowest (E, 0.005 amol) concentrations are indicated with red and green lines in the plot. (**B**) microRNA RPM values are shown together with median (dark blue line) and interquartile range (light blue lines). (**C**) microRNA copy numbers per µL of input RNA are shown together with median (dark blue line) and interquartile range (light blue lines). (**D**) microRNA copy numbers per µL of input RNA for a CNS enriched miRNA, miR-124-3p.

**Table 1 ijms-23-01226-t001:** The miND spike-in sequences. A 13-nucleotide core sequence is flanked by four randomized nucleotides the 5′ and 3′ ends.

Oligo	Sequence (5′–3′)	Molar Amount (amol)
I	(N)(N)(N)(N)ACGAUCGGCUCUA(N)(N)(N)(N)	50
K	(N)(N)(N)(N)UGAACGUCCGUAC(N)(N)(N)(N)	10
M	(N)(N)(N)(N)UCUCGCGCGCGUU(N)(N)(N)(N)	2.5
N	(N)(N)(N)(N)CGAGUAAUGAACG(N)(N)(N)(N)	1.5
H	(N)(N)(N)(N)GCUACACACGUCG(N)(N)(N)(N)	0.1
C	(N)(N)(N)(N)UAUUCGCGGUGAC(N)(N)(N)(N)	0.01
E	(N)(N)(N)(N)ACCUCCGUUUACG(N)(N)(N)(N)	0.005

**Table 2 ijms-23-01226-t002:** The median (Q3–Q1) and inter-quartile range (IQR) of microRNA concentrations (molecules/µL) in 1 µL of analysed biofluids. The number of detected microRNAs and the 5 most abundant microRNAs on average for 3 samples per biofluid are indicated.

Biofluid	Sample	Median (Q3–Q1)	IQR	Number of Detected MicroRNAs	The 5 Most Abundant MicroRNAs on Average
Plasma	1	6 (41.3–1.7)	39.6	668	miR-451a, miR-16-5p, miR-486-5p, miR-92a-3p, miR-103a-3p
	2	4.1 (33.6–0.8)	32.8	707	
	3	3.8 (31.2–0.9)	30.3	658	
Serum	1	10.2 (76.7–2.6)	74.1	900	miR-451a, miR-16-5p, miR-92a-3p, miR-486-5p, miR-19b-3p
	2	14.1 (134.6–3.2)	131.4	925	
	3	21.5 (131.1–7.2)	123.9	600	
Synovial Fluid	1	9.9 (30.0–1.2)	57.6	548	miR-21-5p, miR-23a-3p, miR-451a, miR-221-3p, miR-223-3p
	2	14.4 (45.3–1.8)	87.0	530	
	3	6.0 (19.5–0.8)	37.5	552	
Cerebrospinal Fluid	1	2.7 (12.2–0.9)	11.3	387	miR-21-5p, miR-204-5p, miR-145-5p, miR-99a-5p, miR-221-3p
	2	3.8 (16.4–0.9)	15.5	388	
	3	4.5 (23.1–1.5)	21.6	467	

## Data Availability

The datasets generated during the current study are available in the Gene Expression Omnibus repository (GSE189930).

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
