# Peer review of "A MicroRNA Next-Generation-Sequencing Discovery Assay (miND) for Genome-Scale Analysis and Absolute Quantitation of Circulating MicroRNA Biomarkers"

_ijms, 2022, doi:10.3390/ijms23031226_

Round 1

Reviewer 1 Report

Thorough and well-conceived study. Please review sentences on the the following pages for minor grammatical and style errors - 

1) Introduction - lines 82-85. Starting with "Several commercially available kits... detection".

2) Introduction - lines 86 - Starting with "However, the detection..." Placement of the word However incorrect.

3) Figure 3 - Please indicate replicate number in plots B & C.

4) Figure 4  - Please be consistent in how individual plots are indicated if possible .... "(A)" and "(B)" versus "C-", "D-"... etc.

Author Response

Dear Reviewer,

Thank you for taking time to review our manuscript and for your feedback. We corrected the text according to your suggestions:

  • Lines 82-85 - changed “Several commercially available kits enable to analyse circulating microRNAs in various tissues and body compartments. This approach allowed to generate a comprehensive map of the microRNA expression profiles across different sample types and conditions” to “Several commercially available kits enable to analyse circulating microRNAs in various tissues and body compartments. The NGS technology allowed to generate a comprehensive map of the microRNA expression profiles across different sample types and conditions”
  • Line 86 – changed “However, the detection of microRNA by NGS can be affected depending on the technical methods used for library preparation, which will bias the relative abundance of the selected microRNAs across different samples” to “The detection of microRNA by NGS can be affected depending on the technical methods used for library preparation, which will bias the relative abundance of the selected microRNAs across different samples”
  • Figure 3 – Added information on the number of replicates per protocol (3 replicates)
  • Figure 4 – Changed “(A) and (B)” to “A - and B –“

Best Regards,

Author

Reviewer 2 Report

Dear Authors,

I like the concept of miND assay for determining the correct level of microRNAs in different sample types. By using a novel panel of exogenous small RNA spike-in controls definitely improves the experimental setup for microRNA biomarker discovery in the current setting of valid research and as a potential diagnostic tool. A small typo in the introduction line 57, new sentence, microRNA should be MicroRNA. Let's go for the individual disease states of patients and unravel specific biomarkers for specific diseases.

Well done!

Regards, Reviewer

Author Response

Dear Reviewer,

Thank you for taking time to review our manuscript and for your feedback. We corrected the typo in the introduction line 57.

Best Regards,

Author